# Synergistic Anticancer Effect of Glycolysis and Histone Deacetylases Inhibitors in a Glioblastoma Model

**DOI:** 10.3390/biomedicines9121749

**Published:** 2021-11-23

**Authors:** Beata Pająk, Ewelina Siwiak-Niedbalska, Anna Jaśkiewicz, Maja Sołtyka, Rafał Zieliński, Tomasz Domoradzki, Izabela Fokt, Stanisław Skóra, Waldemar Priebe

**Affiliations:** 1Independent Laboratory of Molecular Biology and Genetics, Kaczkowski Military Institute of Hygiene and Epidemiology, Kozielska 4, 01-163 Warsaw, Poland; ewelinasiwiak1@op.pl (E.S.-N.); ancpatrin@gmail.com (A.J.); maja.soltyka@gmail.com (M.S.); domoradzki.tomasz@gmail.com (T.D.); 2Department of Experimental Therapeutics, The University of Texas MD Anderson Cancer Center, 1901 East Rd., Houston, TX 77054, USA; RJZielinski@mdandreson.org (R.Z.); ifokt@mdanderson.org (I.F.); sskora@maandreson.org (S.S.)

**Keywords:** glycolysis, glioblastoma, 2-deoxy-D-glucose, histone deacetylase inhibitors, drug synergy, anticancer therapy

## Abstract

Over the last decade, we have seen tremendous progress in research on 2-deoxy-D-glucose (2-DG) and its analogs. Clinical trials of 2-DG have demonstrated the challenges of using 2-DG as a monotherapy, due to its poor drug-like characteristics, leading researchers to focus on improving its bioavailability to tissue and organs. Novel 2-DG analogs such as WP1122 and others have revived the old concept of glycolysis inhibition as an effective anticancer strategy. Combined with other potent cytotoxic agents, inhibitors of glycolysis could synergistically eliminate cancer cells. We focused our efforts on the development of new combinations of anticancer agents coupled with 2-DG and its derivatives, targeting glioblastoma, which is in desperate need of novel approaches and therapeutic options and is particularly suited to glycolysis inhibition, due to its reliance on aerobic glycolysis. Herein, we present evidence that a combined treatment of 2-DG analogs and modulation of histone deacetylases (HDAC) activity via HDAC inhibitors (sodium butyrate and sodium valproate) exerts synergistic cytotoxic effects in glioblastoma U-87 and U-251 cells and represents a promising therapeutic strategy.

## 1. Introduction

Glioblastoma (GBM) is the most common and aggressive form of primary brain cancer. Despite optimized therapy, consisting of surgery, chemotherapy, and radiation, its expected median survival remains under two years [1]. Significant progress in understanding the pathogenesis of the disease in recent decades has not been translated into an improved prognosis for patients diagnosed with GBM, which remains dismal [2]. Historically, treatment consisted of maximal surgical resection with radiotherapy alone and had a median overall survival time of approximately 12.1 months. Temozolomide (TMZ), an alkylating agent, is the only first-line drug in GBM therapy and was approved in 2005 [3]. The addition of adjuvant (TMZ) to maximal surgical resection and radiotherapy (Stupp protocol) showed a modest increase in median overall survival times, to 14.6 months [4]. When low-intensity alternating tumor-treating electrical fields (Optune^®^ therapy, Novocure Ltd., Haifa, Israel) are combined with maximal surgical resection, followed by the Stupp regimen, the median overall survival time is prolonged to 20.9 months [5]. However, there is still an unmet medical need for an effective anticancer GBM therapy.

Specific drug delivery to the brain is challenging because of the blood–brain barrier (BBB), which only allows transport through endogenous BBB transporters [6]. The available pharmaceuticals, mostly including small molecules, cannot cross the BBB.

GBM cells are characterized by changes in molecular pathways that fuel their rapid growth and aggressive phenotype [7]. Specifically, cancer cells, including GBM, rely on aerobic glycolysis (the Warburg effect), reducing the tricarboxylic acid (TCA) cycle [8]. This metabolic divergence between GBM and normal cells creates a targeting opportunity that can be explored to develop novel therapeutic strategies.

One of the best-known glycolysis inhibitors is a synthetic analog of glucose: 2-deoxy-D-glucose (2-DG), in which the 2-hydroxyl group is replaced by hydrogen [9]. Similar to D-glucose, 2-DG is transported across the BBB and is quickly taken up by glucose transporters (GLUTs), mainly GLUT1 and GLUT4. Active transport via sodium-glucose co-transporters (SGLT), such as SGLT1, also occurs [10]. Once inside the cells, 2-DG is phosphorylated to 2-deoxy-D-glucose-6-phosphate (2-DG-6-P). A lack of 2-OH group prevents its isomerization to fructose-6-P, and 2-DG-6-P is trapped inside the cell, leading to allosteric and competitive inhibition of hexokinase (HK) and phosphoglucose isomerase (PGI). Inhibition of HK and PGI enzymes causes glycolysis inhibition, ATP depletion, cell cycle arrest, growth inhibition, and cell death [11,12]. As 2-DG is relatively non-toxic and orally available, it is an attractive tool for potential therapies and, thus, has been tested in several clinical trials [9]. It seems that its most promising application may be as a synergistic agent in combination with other cytotoxic compounds. 

The diagnostic potential of 2-DG has been widely explored (^18^F-DG), but the clinical anticancer activity has yet to be established. The main obstacle to an efficient therapy with 2-DG is its rapid metabolism and short half-life (according to Hansen et al., after infusion of 50 mg/kg 2-DG, its plasma half-life was only 48 min.) [13]. Moreover, 2-DG has to be used at relatively high concentrations (≥5 mmol/L) to compete with blood glucose [14]. The adverse effects of 2-DG administration in humans were rather mild and included sweating, dizziness, and nausea, mimicking the symptoms of hypoglycemia [15]. 

To overcome the problems mentioned above and improve 2-DG’s pharmacokinetics and drug-like properties, novel acetylated analogs of 2-DG have been developed [16]. Among the acetylated derivatives, lead compound WP1122 (3,6-di-O-acetyl-2-deoxy-D-glucose) has been selected for further studies [16]. WP1122 enters cells and crosses the BBB by passive diffusion, rather than relying upon GLUTs. Inside the cells, WP1122 undergoes deacetylation by esterases and releases biologically active 2-DG; afterwards, phosphorylation at the C-6 hydroxyl group to 2-DG-6-P inhibits the glycolysis pathway [16]. The ability of WP1122 to target the brain makes it a serious candidate for the treatment of GBM. It was shown that WP1122 demonstrates good oral bioavailability, resulting in a two-fold higher plasma concentration of 2-DG than that achieved via administration of 2-DG alone [16]. Moreover, in vitro experiments confirmed that WP1122 potently inhibits glycolysis in U87 cell lines, revealed by the extracellular acidification rate (ECAR), resulting in 2–10 times more potent anticancer activity, determined by real-time monitoring of the extracellular acidification rate (ECAR); resulting in 2–10 times more potent anticancer activity when compared to 2-DG (half maximal inhibitory concentration (IC_50_) range of 1–10 mM), in both hypoxic and normoxic conditions [16]. Recently, Keith et al. [17] demonstrated a significant real-time inhibition of glycolysis by WP1122 in an orthotopic brain tumor model. 

Despite the metabolic shift, epigenetic abnormalities, including histone acetylation alternations, are also associated with cancer development, including GBM [18]. In general, histone acetylation is associated with chromatin relaxation, given that it neutralizes the positive charge of lysine residues [19]. On the other hand, deacetylation leads to chromatin condensation, creating a structure called heterochromatin. In heterochromatin areas, transcription is repressed. It was shown that in tumorigenesis, the finely tuned acetylation status at the whole proteome level is significantly impaired by dysregulated deacetylases, and histone deacetylases (HDAC) levels are increased [20,21]. In general, the HDAC blockade inhibits tumor growth and induces apoptosis of cancer cells, whereas normal tissue is not particularly affected. HDAC inhibitors (HDACi) can decompose and condense, not only the histone–DNA complex, but also the acetylation status of non-histone proteins [19]. Several clinical trials using HDACi have been performed, and their results indicate that HDACi have antitumor activity [22,23]. For example, suberanilohydroxamic acid (SAHA)/vorinostat, romidepsin, and belinostat were approved by the US Food and Drug Administration (FDA) for the treatment of cutaneous T-cell lymphoma and peripheral T-cell lymphoma [24,25,26]. In addition, panobinostat has demonstrated clinical success and has been approved to treat multiple myeloma [27]. Another HDACi, sodium phenylbutyrate (4-PB), though not in oncology, is approved by the FDA to treat urea cycle disorders and is now being tested for therapy in multiple types of cancer [28]. In addition, a new HDAC inhibitor, CG-745, has been recently granted orphan drug designation by the FDA for pancreatic cancer [29]. In general, more than six hundred HDACi, with more than three hundred and fifty clinical trials completed or being recruited, both as single agents and/or in combinations with other therapeutics, have been tested so far. 

Concerning GBM, it has been shown that the HDAC functions are also altered and determine the abnormal activation of receptor tyrosine kinase (RTK)/Ras/phosphoinositide-3-kinase (PI3K), p53, retinoblastoma (Rb), epidermal growth factor receptor (EGFR), and phosphatase and tensin homolog (PTEN) signaling pathways [30]. Thus, considerable interest in the treatment of GBM using HDACi has been evoked. Several HDACi were reported to penetrate the BBB and exhibit potent activity, alone or when combined with other cytotoxic drugs or radiation therapy (RT) [31]. Two of the most potent HDACi against GBM cells are valproic acid (VPA) and sodium butyrate (NaBt) [32,33].

The current paradigm in cancer treatment is combination chemotherapy, with more than one medication simultaneously [34]. Since chemotherapy drugs can affect cancer cells at different points in the cell cycle, using a combination of drugs increases the chance that all cells are eliminated and that resistance is prevented. Targeting GBM cells through different mechanisms lowers the chance of the development of a resistant phenotype. Based on this hypothesis, we evaluated the cytotoxic effect of simultaneous inhibition of glycolysis and HDAC activity, two upregulated pathways driving GBM growth, using 2-DG or WP1122 with sodium valproate (NaVPA) or NaBt. Here, we report the synergistic cytotoxicity of glycolysis and HDAC inhibitors combination, a promising prospect for future GBM therapy strategies.

## 2. Materials and Methods

### 2.1. Reagents

2-deoxy-D-glucose (2-DG), sodium butyrate (NaBt), sodium valproate (NaVPA), bovine serum albumin (BSA), phenazine methosulfate (PMS), sulforhodamine B (SRB), trichloroacetic acid (TCA), acetic acid, dimethyl sulfoxide (DMSO), cycloheximide (CHX), chloroquine (CQ), dimethyloxalylglycine (DMOG), rhodamine (Rho), 5-bromodeoxyuridine (BrdU) Cell Proliferation Assay Kit, and Lactate Assay Kit were purchased from Sigma-Aldrich, Saint Louis, MO, USA. 3,6-di-O-acetyl-2-deoxy-D-glucose (WP1122) was produced at the MD Anderson Cancer Institute by Professor Waldemar Priebe’s team and was kindly provided for research by the Dermin company, who, based on an exclusive license, conduct developmental studies of the molecule. 

3-(4,5-dimethylthiazol-2-yl)-5-(3-carboxymethoxyphenyl)-2-(4-sulfophenyl)-2H tetrazolium, inner salt (MTS), was purchased from Promega (Madison, WI, USA). Muse™ Annexin V & Dead Cell Kit was purchased from Calbiochem, Merck Millipore (Darmstadt, Germany). HDAC Activity Colorimetric Assay Kit was purchased from BioVision (Milpitas, CA, USA). All the other reagents and solvents used in this study were of the highest analytical reagent grade. 

### 2.2. Cell Cultures and Treatments

The human glioblastoma U-87 and U-251 cell lines were obtained from the European Collection of Authenticated Cell Cultures (ECACC, Salisbury, Wiltshire, UK). Cells were cultured in growth media (GM) constituted by Dulbecco’s modified Eagle’s medium (DMEM, Gibco^TM^—Life Technologies, Grand Island, NY, USA) with low (1 g/L for U-87) or high (4.5 g/L for U-251) glucose concentration, supplemented with 10% (*v*/*v*) heat-inactivated fetal bovine serum (FBS, Biowest, Riverside, MO, USA), Penicillin/Streptomycin (Life Technologies/Thermo Fisher Scientific, Waltham, MA, USA; 50 IU/mL/50 μg/mL), Gentamicin sulfate (Sigma-Aldrich, Saint Louis, MO, USA; 20 μg/mL), and Fungizone/Amphotericin B (Thermo Fisher Scientific, Waltham, MA, USA; 1 μg/mL), and grown until 70–80% confluence. Next, cells were seeded in 6-well, 24-well, or 96-well plates or Petri dishes, depending on the type of experimental protocol. 

Treatment of the cultures with the different experimental factors tested in the current work was performed for 48 or 72 h. Experimental compounds were tested at various concentrations: glycolysis inhibitors, 2-DG: 0.5–20 mM, WP1122: 0.25–5 mM; histone deacetylase inhibitors (HDACi), NaBt: 2.5–20 mM, and NaVPA: 2.5–20 mM. For each compound, IC_50_ concentration was calculated. GBM cells were treated with each compound alone or with glycolysis inhibitors + HDACi combinations (2-DG + NaBt, 2-DG + NaVPA, WP1122 + NaBt, WP1122 + NaVPA). As a positive control of the apoptosis inducer, a protein-synthesis inhibitor, cycloheximide (CHX), was used. Due to CHX-induced cell detachment and possible cell number reduction, media from CHX-treated cells were collected, centrifuged, and scraped, and the retrieved cells were also analyzed. For viability, proliferation, and protein synthesis assays, cells were seeded in 96-well plates at 1000 cells/well. To assess the importance of autophagy induction, a chloroquine (CQ)-inhibiting autophagosome–lysosome fusion agent was used.

To assess the influence of hypoxia on the GBM cells’ sensitivity to the tested compounds, hypoxia-mimicking conditions were applied. To mimic hypoxic conditions, the prolyl hydrolase inhibitor (DMOG) (50 or 100 μM) and rhodamine (Rho) (0.25 or 0.5 μM) were added 4 h before the beginning of the experiment and were present in the medium until the end of the incubation with experimental factor (48 and 72 h). To confirm the activation of the HIF-1α pathway and its transcriptional activity, the expression of downstream proteins, such as pyruvate dehydrogenase kinase (PDK1) and lactate dehydrogenase A (LDHA), were examined with the Western blot method.

All experiments were performed in triplicate and repeated at least three times, with similar results. 

### 2.3. Assessment of Cell Viability

Cell viability was assessed by evaluating the ability of cells to convert soluble MTS (3-(4,5-dimethylthiazol-2-yl)-5-(3-carboxymethoxyphenyl)-2-(4-sulfophenyl)-2H-tetrazolium, inner salt) into an insoluble purple formazan (Promega, Madison, WI, USA). Briefly, cells grown and treated with the tested compounds were incubated for 1 h at 37 °C with MTS solution (20 μL per well). Color determination in MTS assay was measured at 490 nm using a Synergy H1 multi-plate reader (BioTek, Winooski, VT, USA). 

### 2.4. Assessment of Cell Proliferation

For the proliferation assay, cells were seeded in a 96-well plate at 1000 cells/well. DNA synthesis in proliferating U-87 and U-251 cells was determined using a BrdU Cell Proliferation Assay Kit, according to the manufacturer’s protocol (Sigma-Aldrich, Saint Louis, MO, USA). Briefly, 20 μL of the diluted 1x BrdU solution was added to the cells for 24 h. Cells were washed with phosphate-buffered saline (PBS) and fixed for 30 min in fixing solution. Next, cells were incubated with mouse anti-BrdU-antibody (1.5 h, Room Temperature (RT)), washed three times with PBS, and subsequently incubated with anti-mouse horseradish peroxidase (HRP) IgG (1:2000) (0.5 h, RT). Finally, HRP substrate (tetramethylbenzidine, TMB) was added for 30 min at RT. The reaction was stopped with Stop solution, and the absorbance at 450 nm was determined with a microplate reader (BioTek). 

### 2.5. Assessment of Protein Synthesis

In brief, a sulforhodamine B (SRB) assay was used to measure cellular protein content. The method described here was optimized for the toxicity screening of compounds to adherent cells in a 96-well format, as described by Orellana and Kasinski [35]. After an experiment was completed, cell monolayers were fixed with 10% (*w*/*v*) trichloroacetic acid and stained for 30 min, after which the excess dye was removed by repeated gentle washing with 1% (vol/vol) acetic acid. The protein-bound dye was dissolved in 10 mM Tris base solution (pH 10.5). Color determination was measured at 510 nm using a Synergy H1 multi-plate reader (BioTek). 

### 2.6. Cell Apoptosis Assay

Cell apoptosis was assayed using a Muse™ Annexin V and Dead Cell Kit (Merck Millipore, Guyancourt, France) according to the user guide. Cells were treated with 3–4 concentrations of tested compounds, covering the IC50 value. After experiments, cells were trypsinized (Gibco^TM^, Life Technologies, Grand Island, NY, USA), collected by centrifugation (3000 rpm, 5 min), and washed with PBS. Cells were resuspended in PBS, mixed with Muse™ Annexin V and Dead Cell reagent, and then incubated for 20 min at RT in the dark. Assay results were measured using a Muse™ Cell Analyzer (Luminex Corporation, Austin, TX, USA). 

### 2.7. Ultrastructural Studies with Transmission Electron Microscopy (TEM)

Cells were fixed in 2% paraformaldehyde (PFA) 2.5% glutaraldehyde in 0.1 M sodium cacodylate buffer (pH 7.4) for 1 h at 4 °C. Cells were washed with the same buffer and post-fixed with 1% OsO_4_ in 0.1 M sodium cacodylate buffer for 1 h, at RT. Cells were dehydrated in a graded ethanol alcohol series and finally in pure propylene oxide (10 min), and then embedded in Epon 812. Ultrathin sections were mounted on copper grids, air-dried, and stained for 10 min with 4.7% uranyl acetate and for 2 min with lead citrate. The sections were examined and photographed with a JEOL JEM 1400 (JEOL Co., Tokyo, Japan) electron microscope. TEM analyses were performed in the Laboratory of Electron Microscopy, Nencki Institute of Experimental Biology of Polish Academy of Sciences, Warsaw, Poland. 

### 2.8. Preparation of Whole-Cell Ly Sates

Cells were grown on 100-mm diameter culture Petri dishes. To obtain whole-cell lysates, a 1-mL aliquot of ice-cold PBS was added, and cells were immediately scraped from the plastics and collected by centrifugation (10,000× *g* for 10 min, 4 °C). A 1.0 mL aliquot of radioimmunoprecipitation assay (RIPA) buffer (1x PBS, 10 mL/L Igepal CA-630, 5 g/L sodium deoxycholate, 1 g/L SDS), supplemented with 0.4 msM phenylmethylsulfonyl fluoride (PMSF), 10 μg/mL of aprotinin, and 10 μg/mL of sodium orthovanadate (Sigma-Aldrich, St. Louis, MO, USA), was added to lyse the cell pellet, and cells were broken up by repetitive triturating with a syringe with attached needle (0.6 mm diameter). The cell suspension was then left on ice (4 °C) for 30 min, then centrifuged for another 5 min (4 °C, 10,000× *g*). Next, the viscous solution was divided into smaller volumes and transferred to fresh Eppendorf tubes and stored at −80 °C until use.

### 2.9. Preparation of Cytoplasmic and Nuclear Lysates

Cells were grown on 100-mm diameter culture Petri dishes. Following each experiment, the cells were washed twice with PBS, scraped off in PBS, and centrifuged (4 °C, 10,000× *g*, 5 min). Cell pellets were stored at −80 °C to the end of the experiment. Cell pellets were resuspended in 400 μL of ice-cold buffer (10 mM 4-(2-hydroxyethyl)piperazine-1-ethanesulfonic acid (HEPES) pH 7.9; 10 mM KCl; 0.1 mM ethylenediaminetetraacetic acid (EDTA); 0.1 mM egtazic acid (EGTA); 1 mM 1,4-dithiothreitol (DTT); 0.5 mM PMSF), then incubated on ice for 15 min, after which 25 μL of a 100 mL/L solution of Igepal CA-630 was added. After centrifugation (20 °C, 1000× *g*, 30 s), supernatants containing cytoplasm were transferred to fresh tubes and stored at −20 °C. Nuclear pellets were resuspended in 200 μL RIPA buffer, as described for whole-cell lysates. PMSF (0.1 mg/mL) was added, and cells were incubated for 30 min on ice. After centrifugation (4 °C, 10,000× *g*, 5 min) nuclear lysates were stored at −80 °C until analysis. Soluble protein concentrations in the whole cell, the cytoplasmic, and the nuclear fractions were determined using a protein-dye-binding method [36] with commercial Bradford reagent (Sigma-Aldrich, Saint Louis, MO, USA).

### 2.10. Preparation of Whole-Cell Lysates

Each protein extract was adjusted to a 2 μg/μL concentration with the addition of Laemmli sample buffer (4X concentrate, Bio-Rad, Hercules, CA, USA), heated for 5 min at 95 °C, and separated on precast polyacrylamide gradient gels (7.5–15%). After transfer to 0.2 μm polyvinylidene difluoride membranes (PVDF, Bio-Rad), the membranes were blocked in either 5% (*w*/*v*) nonfat dried milk or 1% (*w*/*v*) bovine serum albumin (BSA) (Sigma-Aldrich, Saint Louis, MO, USA) diluted in Tris-buffered saline containing 0.1% (*v*/*v*) Tween 20 (TBS-T), and probed overnight at 4 °C with the respective primary antibodies, as follows: mouse anti-actin (1:2500), rabbit anti-LDHA (1:1000), rabbit anti-microtubule-associated protein 1B/2B light chains 3B (MAP-LC3) (1:1000) (Cell Signaling Technology, Danvers, MA, USA), rabbit anti-HIF-1α (1:250), rabbit anti-PDK1 (1:1000), rabbit anti-Bax (1:1000), rabbit anti-Bcl-2 (1:500), and rabbit anti-caspase 3 (1:250) (Abcam, Cambridge, UK). Membranes were subsequently incubated with the respective HRP-linked anti-rabbit or anti-mouse IgG (1:1000) (Cell Signaling Technology, Danvers, MA, USA) antibodies and developed using ECL or ECL Plus (WesternBright Sirius Chemiluminescent Detection Kit, Advansta Inc., USA). Detection and densitometric quantification of band intensities was performed using a G-Box system (Syngene, Cambridge, UK). 

### 2.11. Lactate Quantification

To assess the intensity of the glycolysis process, intra- and extracellular levels of lactate were quantified using a colorimetric Lactate Assay Kit (Sigma-Aldrich, Saint Louis, MO, USA) according to the manufacturer’s protocol. Briefly, cells were grown on 60-mm diameter culture Petri dishes. At the end of experiments, cells were washed with ice-cold PBS, scraped off, and centrifuged (4 °C, 10,000× *g*, 6 min). Cell pellets were resuspended in 500 μL of ice-cold PBS and centrifuged again (4 °C, 10,000× *g*, 6 min). To remove intracellular lactate dehydrogenase, cell pellets were resuspended in 500 μL of Lactate Assay Buffer, transferred to Pierce Protein Concentrators with polyethersulfone (PES) (Thermo Fisher Scientific Invitrogen, Waltham, MA, USA), and centrifuged (4 °C, 13,000× *g*, 10 min). The obtained soluble fractions were used for intracellular lactate quantification. A total of 50 μL of cell supernatant was transferred to 96-well plate in triplicates. To evaluate the extracellular lactate level, after the end of the experiments, 50 μL of cell supernatant (culture medium) was diluted and transferred to 96-well plate in triplicates. Next, 50 μL of Master Reaction Mix containing 46 μL Lactate Assay Buffer + 2 μL Lactate Enzyme Mix + 2 μL Lactate Probe was added, and plates were incubated for 15 min in the dark. Color determination was performed at 570 nm using a Synergy H1 multi-plate reader (BioTek). 

### 2.12. HDAC Activity Assay

HDAC activity was assessed with a commercial colorimetric assay (BioVision, Milpitas, CA, USA). Briefly, after the end of the experiment, whole-cell protein lysates were prepared according to a previously described method, and protein concentration was assessed. A total of 100 μg of protein lysates was incubated for 90 min at 37 °C with 10 μL HDAC Assay Buffer and 5 μL of HDAC colorimetric substrate. The reaction was stopped by adding 10 µL of Lysine Developer. Next, samples were incubated for 30 min at 37 °C, and color determination was performed at 400 nm with a Synergy H1 multi-plate reader (BioTek). As a positive control, cell lysate from control, untreated cells with 2 μL of Trichostatin A (TSA), a potent HDAC inhibitor, was used.

### 2.13. Drugs Interaction Analysis

To assess the mechanism of drug interaction, the viability of cells treated with different proportions of the above-mentioned drugs (1:10, 1:5, 1:2.5, and 1.25) was subjected to statistical analysis using the Chou–Talalay method [37]. This method allows determining the so-called drug combination index (CI), where a CI < 0.1 signifies a very strong synergy; CI 0.1–0.3, strong synergy; CI 0.31–0.7, synergy; CI 0.71–0.85, moderate synergy; CI 0.86–0.9, slight synergy; CI 0.91–1.10, nearly additive; CI 1.11–1.20, slight antagonism; CI 1.21–1.45, moderate antagonism; CI 1.46–3.30, antagonism; CI 3.31–10,very strong antagonism; and CI > 10, very strong antagonism. Results were analyzed with CompuSyn software version 1.0 (ComboSyn Inc., Paramus, NJ, USA)

### 2.14. Statistical Analysis

The results were analyzed with one-way ANOVA test, with Tukey’s multiple comparisons and/or two-way ANOVA test, followed by Bonferroni’s multiple comparisons using GraphPad PrismTM version 8.0 software (GraphPad Software Inc., San Diego, CA, USA). Error bars indicate the standard error of mean (S.E.M.), and significant differences are indicated as follows: * *p* < 0.05, ** *p* < 0.01, *** *p* < 0.001 for comparison between the means. Results were obtained as means of at least three independent experiments.

## 3. Results

### 3.1. Glycolysis Inhibitors Exert a Cytotoxic Effect on GBM Cells

Examination of U-87 and U-251 GBM cells revealed that the 2-DG and WP1122 treatments dose- and time-dependently exerted cytotoxic effects, as examined with the MTS viability (Figure 1), BrdU proliferation (Figure 2), and SRB protein synthesis (Figure 3) assays. CHX (20 µM), a known protein-synthesis inhibitor, was used as a positive control for the cytotoxic effect. Additionally, IC50 values were calculated: for U-87 cells: 2-DG: 20 mM (48 h) and 5 mM (72 h), WP1122, 3 mM (48 h) and 2 mM (72 h); and U-251 cells: 2-DG: 12 mM (48 h), 5 mM (72 h), WP1122: 1.25 mM (48 h) and 0.8 mM (72 h). It is clear that acetylation significantly improved the WP1122 properties, and its effective concentrations were lower than 2-DG, raising the possibility of the compound’s clinical application.

It is postulated that hypoxic conditions make GBM cells almost completely dependent on the glycolysis process. Thus, we assumed that in hypoxia-like conditions, the cytotoxic action of glycolysis inhibitors would exert a more potent effect than under normoxia. To verify this hypothesis, hypoxia-like conditions were induced using DMOG (50 and 100 μM) and rhodamine (Rho) (0.25 and 0.5 μM), alone or in combination, in non-toxic concentrations (Appendix A). Hypoxia-like state induction was confirmed by evaluating HIF-1α protein expression and its downstream targets: PDK1 and LDHA (Appendix A). As expected, the levels of the detected proteins were upregulated in response to DMOG + Rho treatment (Appendix A).

Interestingly, in hypoxia-like conditions, WP1122 cytotoxicity was not significantly different from normoxia treatment (Appendix A). Moreover, in U-87 cells, the cytotoxicity of 2-DG was even significantly lower than in a normoxia state (Appendix A). These results indicate the highly glycolytic phenotype of GBM cells (Appendix A), even in normoxia conditions. This observation agrees with the well-known fact that GBM and pancreatic cancer cells represent the most glycolytic tumors.

### 3.2. 2-DG and WP1122 Inhibits Lactate Production

To verify the efficiency of glycolysis inhibition, the production of lactate in cell lysates, as well as culture supernatants, was examined using a Lactate Assay Kit (Sigma-Aldrich, Saint Louis, MO, USA), after 72 h of 2-DG (2.5–10 mM) and WP1122 (1–5 mM) treatment. We found that both compounds dose-dependently downregulated the level of lactate in comparison to the untreated cells (Figure 4), indicating a reduction in glucose utilization in the glycolysis pathway.

### 3.3. 2-DG and WP1122 Cytotoxicity Is Mediated via Apoptosis Induction

To verify the involvement of apoptotic cell death in 2-DG- and WP1122-induced GBM cell cytotoxicity, an Annexin V Cell Death Kit was used. As shown in Figure 5, the viability, proliferation, and protein synthesis reduction corresponded with apoptosis induction in a dose-dependent manner.

### 3.4. HDAC Inhibitors Are Cytotoxic to GBM Cells

As mentioned previously, the HDAC activity in GBM cells is significantly altered. Thus, we used the well-known HDAC inhibitors, sodium butyrate (NaBt) and sodium valproate (NaVPA), to verify their anticancer action. As shown in Figure 6, Figure 7 and Figure 8, NaBt (2.5–20 mM) and NaVPA (2.5–20 mM) dose-dependently reduced the U-87 and U-251 cell viability (Figure 6), proliferation (Figure 7), and protein synthesis (Figure 8). The calculated IC50 values were as follows: U-87 cells: NaBt, 14 mM (48 h) and 10 mM (72 h); NaVPA, 15 mM (48 h) and 10 mM (72 h); and U-251 cells: NaBt, 15 mM (48 h), 10 mM (72 h); NaVPA, 15 mM (48 h) and 12.5 mM (72 h). Similarly to the 2-DG and WP1122 analysis, the susceptibility of U-87 and U-251 cells to HDACi action was also examined under hypoxia-like conditions. As shown in Appendix A, the cytotoxicity under hypoxia-like conditions was not significantly different from the normoxia treatment.

### 3.5. HDACi Cytotoxicity Is Mediated via Apoptosis Induction

To verify the involvement of apoptotic cell death in NaBt- and NaVPA-induced GBM cell cytotoxicity, an Annexin V Cell Death Kit was used. As shown in Figure 9, the viability, proliferation, and protein synthesis reduction corresponded with apoptosis induction in a dose-dependent manner.

### 3.6. HDAC Inhibitors Modulate HDAC Enzymes’ Activity and Regulate Pro- and Antiapoptotic Genes Expression

To confirm the molecular mechanism of NaBt and NaVPA action, the HDAC enzyme activity was assessed using an HDAC Activity Colorimetric Assay Kit (BioVision). As shown in Figure 10, NaBt (5–15 mM) and NaVPA (5–15 mM) inhibited the HDAC activity in a dose-dependent manner, leading to changes in gene transcription.

The altered activity of HDAC resulted in a modified expression of anti- and proapoptotic proteins, facilitating apoptosis execution. As shown on Figure 11, an IC50 concentration of NaBt (10 mM for U-87 and U-251 cells) and NaVPA (10 mM for U-87 and 12.5 mM for U-251 cells) (72 h) downregulated the expression of the antiapoptotic Bcl-2 protein. At the same time, in U-87 cells, NaBt and NaVPA treatments significantly induced proapoptotic Bad and Bax protein levels. Similarly, in U-251 cells, NaVPA increased Bad and Bax protein levels, whereas NaBt treatment only affected the Bax protein expression (Figure 11).

### 3.7. Glycolysis- and HDAC-Inhibitor Treatments Activate the Protective Autophagy Process

Autophagy is a crucial cellular mechanism, activated in response to various stimuli. In cancer cells, autophagy plays dual roles in tumor promotion and suppression and contributes to cancer-cell development and proliferation, depending on the cancer type and stage. We assumed that glycolysis inhibition via 2-DG and WP1122 and impaired glucose utilization could initiate autophagy as an alternative source of energy production. Furthermore, autophagy may have supported the cytotoxic action of all the tested compounds. To assess the role of autophagy in 2-DG, WP1122, NaBt, and NaVPA cytotoxicity, chloroquine (CQ), in a non-toxic concentration of 10 μM (Appendix A), was used. CQ is a classic autophagy inhibitor that blocks the binding of autophagosomes to lysosomes by altering the acidic environment of the lysosome, resulting in the accumulation of a large number of degraded proteins in cells [38]. Thus, CQ treatment results in MAP-LC3 protein accumulation within the cell (a so-called autophagic flux). 

Surprisingly, we found that autophagy is not a critical process for the cytotoxic action of the analyzed compounds. Its inhibition with CQ did not significantly modulate the U-87 and U-251 cell viability when treated with IC50 concentrations of 2-DG, WP1122, NaBt, and NaVPA for 72 h (Appendix A). Furthermore, MAP-LC3 expression was not significantly altered after the treatments (Figure 12). On the other hand, the presence of CQ (10 μM) resulted in MAP-LC3 accumulation, confirming the physiological autophagy process within GBM cells. 

Interestingly, a representative analysis of cell ultrastructure with TEM showed the presence of numerous autophagic vacuoles (AV), especially autophagolysosomes, in U-87 cells treated with 2-DG (5 mM, 24 h) and WP1122 (2 mM, 24 h) (Figure 13). On the other hand, in NaBt (10 mM)-and NaVPA (10 mM)-treated cells, autophagolysosomes were not visible. This suggests that glycolysis inhibition can modulate the autophagy process, but its downstream cellular effects and lack of ATP synthesis strongly induced autophagy-independent cell death. It is also possible that autophagy is a primary response to glycolysis inhibition and is activated very early; thus, it was observed with the TEM analysis after 12 h of incubation, whereas viability was assessed after 48 and 72 h of treatment. It should also be noted that in 2-DG- and WP1122-treated cells, the mitochondria morphology is also affected, and their significant condensation, with loss of cristae structure, is observed (Figure 13). Further studies are needed to elucidate the exact molecular mechanism of 2-DG and WP-1122’s influence on mitochondrial function in GBM cells.

### 3.8. Glycolysis and HDAC Inhibitors Synergistically Eliminate GBM Cells

According to the presented data, by targeting cellular metabolism via glycolysis inhibitors (2-DG, WP1122) and modulating cellular transcription processes with HDAC inhibitors (NaBt, NaVPA), we can eliminate GBM cells. Our concern, however, is raising the effective concentrations of the tested compounds, which are relatively high and could be difficult to enrich in human organisms. Furthermore, current anticancer therapies are mainly based on combined treatments, rather than a monotherapy. Thus, we verified whether the concomitant administration of glycolysis and HDAC inhibitors could potentiate their cytotoxic effects. As shown in Figure 14, NaBt and NaVPA significantly potentiated 2-DG (Figure 14A,B) and WP1122 (Figure 14C,D) cytotoxic action, represented by reduced cell viability. In the case of WP1122, U-251 cells were more sensitive to combined treatments, whereas, in U-87 cells, the viability of WP1122 + NaBt/NaVPA-treated cells was not significantly different from WP1122 alone (Figure 14C,D).

To verify the mechanism of drug interactions, the viability of cells treated with different proportions of the drugs mentioned above (1:10, 1:5, 1:2.5, and 1.25) was subjected to statistical analysis using the Chou–Talalay method [37]. This method allows determining the so-called drug combination index (CI), where: CI <0.1 signifies a very strong synergy; CI 0.1–0.3, strong synergy; CI 0.31–0.7, synergy; CI 0.71–0.85, moderate synergy; CI 0.86–0.9, slight synergy; CI 0.91–1.10, nearly additive; CI 1.11–1.20, slight antagonism; CI 1.21–1.45, moderate antagonism; CI 1.46–3.30, antagonism; CI 3.31–10, very strong antagonism; and CI > 10, very strong antagonism. As shown in Appendix A, 2-DG or WP1122 combined treatments with NaBt or NaVPA could exert various synergistic effects—from strong synergy to moderate synergy, depending on the drug–drug proportions—making it a promising therapeutic strategy.

## 4. Discussion

Recently, glycolysis inhibitors have been intensively studied as a group of compounds for potential anticancer therapy [9]. Otto Warburg discovered that most cancer cells are characterized by increased glucose uptake, up to 10-fold higher than that of normal cells [39]. Cancer cells undergo a metabolic switch from oxidative phosphorylation (OXPHOS) to glycolysis, in which a molecule of glucose is degraded to two molecules of pyruvate [40]. Depending on the supply of oxygen to the cells, pyruvate is either reduced to lactate in the absence of oxygen via an anaerobic glycolysis pathway or oxidized to yield acetyl-coenzyme A in the presence of oxygen, and then oxidized completely to CO_2_ and H_2_O via the citric acid cycle [40]. The majority of cancer cells, including GBM, depend on high rates of glycolysis for growth and survival, even when there is sufficient oxygen [41]. This type of aerobic glycolysis is called the Warburg effect. The Warburg effect has long been linked to hypoxia, but it also occurs under normoxic conditions [42]. This statement corresponds to our observations that GBM cells are highly dependent on the glycolysis process, even in normoxia. The use of chemical compounds mimicking hypoxia-like conditions did not modulate the cytotoxicity of the tested glycolysis inhibitors (2-DG and WP1122, Appendix A). 

2-DG is a synthetic glucose analog, in which the 2-hydroxyl group is replaced by hydrogen. Like D-glucose, 2-DG is transported across the BBB and quickly taken up by the cells, mainly by glucose transporters (facilitated diffusion) [10]. Once inside the cells, 2-DG is phosphorylated to 2-deoxy-d-glucose-6-phosphate (2-DG-6-P), a charged compound trapped inside the cell. 2-DG-6-P cannot undergo further isomerization to fructose-6-P, due to the lack of -OH group, and, thus, accumulates inside cells. Similarly to in our studies, the cytotoxicity of 2-DG against various cancer models, including GBM, was previously confirmed [1,9,43]. As 2-DG is relatively non-toxic and orally available, it is an attractive tool for potential therapies [44]. It seems that its most promising application may be as a synergistic agent in combination. 2-DG has been explored/tested as an adjuvant agent for various groups of clinically used chemotherapeutic drugs (such as cisplatin, doxorubicin, daunorubicin, gemcitabine, sorafenib, adriamycin, and others) in breast, prostate, ovarian, lung, glioma, and other cancer types [9]. Despite the numerous preclinical and clinical studies, the use of 2-DG in anticancer therapy is still limited. Its rapid metabolism and short half-life (according to Hansen et al., after infusion of 50 mg/kg 2-DG, its plasma half-life was only 48 min) [13], making 2-DG a rather poor drug candidate. Moreover, 2-DG has to be used at a relatively high concentration (5 mmol/L) to compete with blood glucose. 

To improve 2-DG’s pharmacokinetics and drug-like properties, novel analogs of 2-DG were synthesized, and potential prodrugs were prepared and tested.

One of the most highly promising groups of compounds, acetyl 2-DG analogs, have been developed in Dr. Waldemar Priebe’s laboratory. Among the tested derivatives, lead compound WP1122 (3,6-di-O-acetyl-2-deoxy-d-glucose) has been selected for further studies [16]. Due to its lipophilic properties, related to the presence of the acetyl groups, WP1122 enters cells and, importantly, crosses the BBB by passive diffusion, rather than relying upon a specific glucose transporter. It undergoes deacetylation by intracellular esterases, releasing an active 2-DG molecule trapped inside the cell after phosphorylation at the C-6 hydroxyl group, as described above for 2-DG. The ability of WP1122 to target brain tissue makes it a serious candidate for the treatment of brain tumors, including GBM. It has been shown that, whereas 2-DG is rapidly metabolized, the prodrug WP1122 releases 2-DG, increasing its half-life and its therapeutic effect [45]. These observations are confirmed by the presented results, showing that a IC50 (72 h) concentration of WP1122 in GBM cell line models (U-87 and U-251) was notably lower (2 mM for U-87 and 0.8 mM for U-251 cells) in comparison to 2-DG alone (5 mM for both cell lines). WP1122, similarly to 2-DG, exerted a cytotoxic effect on GBM cells, confirmed by the reduced viability (Figure 1), proliferation (Figure 2), protein synthesis (Figure 3), and lactate production (Figure 4) in U-87 and U-251 cells. At the same time, activation of apoptosis was observed (Figure 5). Furthermore, based on TEM cell ultrastructure analysis (24 h), we observed the upregulation of the autophagy process, visualized by the numerous characteristic autophagic vacuoles—mainly late autophagolysosomes, in 2-DG and WP1122-treated U-87 cells—which were not detected in control cells (Figure 13). However, autophagy did not appear in 2-DG- and WP1122-treated cells as a cell death pathway, because its inhibition with an autophagy inhibitor, chloroquine, did not protect the cell viability (72 h) from 2-DG and WP1122 action (Appendix A). We hypothesize that in our GBM model, autophagy could be a protective mechanism against the glycolysis inhibitor action, being activated in the early phase of their action. However, after prolonged exposure to 2-DG and WP1122 (72 h), a lack of ATP synthesis and possibly other intracellular 2-DG-dependent effects that were not examined in this study (such as unfolded protein response (UPR) stress, reactive oxygen species (ROS) generation, or others) strongly disrupted cellular processes, leading to apoptotic cell death. It should be underlined that WP1122 demonstrated good oral bioavailability, resulting in a two-fold higher plasma concentration of 2-DG than that achieved via administration of 2-DG alone [16]; thus, its clinical application is highly promising. 

Regardless of metabolic reprogramming, GBM cells are characterized by deregulated epigenetic mechanisms, resulting from the aberrant activity of histone deacetylases (HDAC), which remove acetyl groups from the histones regulating chromatin accessibility [46]. HDACis are currently being used in numerous clinical trials as anticancer agents or adjuvant compounds. Furthermore, HDACis have pleiotropic cellular effects and induce the expression of proapoptotic proteins, facilitating cancer cells’ elimination [47,48]. Our studies confirmed that NaBt and NaVPA inhibited the HDAC activity (Figure 10), leading to upregulated proapoptotic Bad and Bax proteins levels, with a concomitant downregulation of antiapoptotic Bcl-2 protein in U-87 and U-251 GBM cell lines (Figure 11). Consequently, the cell viability (Figure 6), proliferation (Figure 7), and protein synthesis (Figure 8) were reduced, leading to intrinsic apoptosis induction (Figure 9). Cytotoxic action of NaBt and NaVPA has been previously reported in GBM models, including U-87 [49] and U-251 cells [50] and in vivo models [51]. Interestingly, Nguyen et al. [52] recently showed that the use of HDAC inhibitors such as Panobinostat, Vorinostat, and Romidepsin downregulated c-Myc expression and glycolytic enzymes, but upregulated OXPHOS in a GBM model. Furthermore, HDACi modulated the pro- and antiapoptotic Bcl-2 family proteins, facilitating intrinsic apoptosis induction [52]. 

The majority of HDACi’s beneficial anticancer actions are based on combined treatments. For example, Kim et al. [53] showed that NaBt was able to sensitize glioma cells to TNF-related, apoptosis-inducing ligand (TRAIL)-induced apoptosis via downregulation of antiapoptotic XIAP and survivin proteins [53]. On the other hand, the efficacy of temozolomide action was potentiated by the presence of VPA, which downregulated the expression of the O6-methylguanine-DNA methyltransferase (MGMT) enzyme, which plays an important role in cellular resistance to alkylating agents [50]. Additionally, the radio-sensitivity of glioma U-251 cells increased, due to the NaBt-induced reduction in the Ku70 protein and an increase in the y-H2AX foci, which decreased the ability for double-strand break (DSB) repair [54]. 

The amalgamation of various anticancer compounds targeting distinct key cellular pathways enhances their efficacy compared to monotherapies, ideally in a synergistic manner. Thus, we verified whether the simultaneous inhibition of glycolysis and HDAC activity could potentiate their cytotoxic effects in a GBM model. We found that 2-DG or WP1122 combined with NaBt or NaVPA were able to eliminate GBM cells synergistically. As far as we know, this is the first evidence that targeting the GBM metabolism along with gene transcription processes is an effective anticancer strategy. It should be underlined that the presented results were obtained from in vitro studies and should be further confirmed in in vivo models. We are also aware that clinical approval of combined therapy is more time- and money-consuming. Thus, the potential of using two new molecules is a future clinical perspective. Based on the described results, Prof. Priebe’s group synthesized a new 2-DG analog, WP1234 (3,6-di-O-(2-Ethyl)butyryl-D-glucose), containing an ethyl-butyrate group in its structure. After entering cancer cells, WP1234 undergoes enzymatic cleavage, releasing active 2-DG and butyryl derivatives, thus simultaneously modulating glycolysis and HDAC action. Its anticancer properties in the GBM model are currently under scrutiny in our laboratory and will be published soon. 

## 5. Perspectives

The complexity of cancer underscores the need for various treatment approaches; therefore, a combination of one or more therapeutic interventions is often used to battle cancer. Combination therapy might achieve efficacy with lower doses or less-toxic drugs. It can chemo-sensitize cells, making an additional compound more potent. It can have additive or synergistic effects, minimize drug resistance, or fight against expected resistance. Although monotherapies are still a common approach, it is often deemed that even better results could be obtained when these therapies are rationally combined with others. Our studies proved that combining glycolysis inhibitors and HDAC inhibitors could more efficiently induce GBM cells apoptosis than a monotherapy. We found that WP1122, a diacetyl 2-DG derivative, is a potent cytotoxic agent and a promising candidate to be a clinically approved drug. Furthermore, valproic acid is an already-approved drug used alone or with other medications to treat certain types of seizures. Thus, its potential use in other indications is feasible. We believe that targeting glycolysis and HDAC activity is a rational strategy for anti-GBM therapy; therefore, this will be further explored by our team.

## Figures and Tables

**Figure 1 biomedicines-09-01749-f001:**
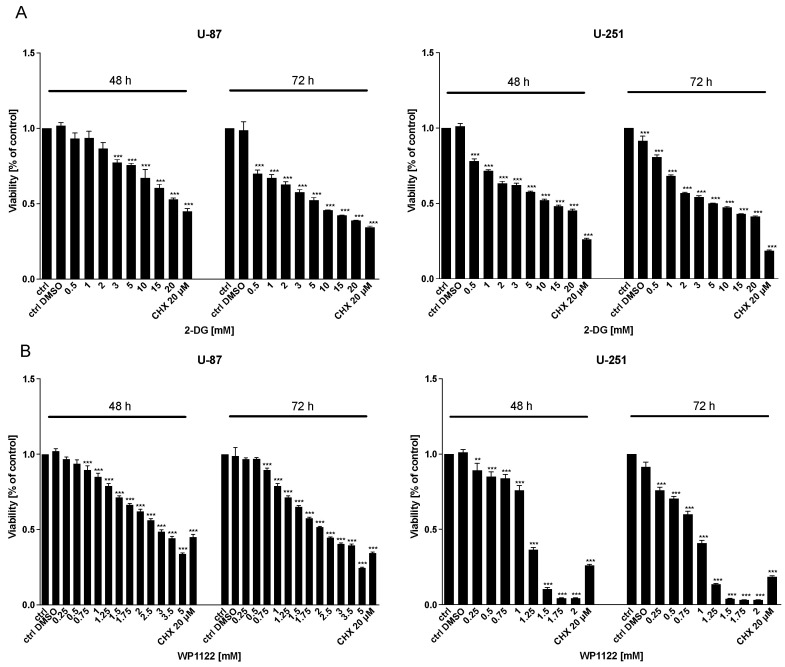
MTS assay showing the viability of U-87 and U-251 cells after 48 and 72 h of treatment with various concentrations of (**A**) 2-DG (0.5–20 mM) and (**B**) WP1122 (0.25–5 mM). CHX (20 μM) was used as a positive control. Significant differences between the treatment means and control value are indicated by ** *p* < 0.01 and *** *p* < 0.001.

**Figure 2 biomedicines-09-01749-f002:**
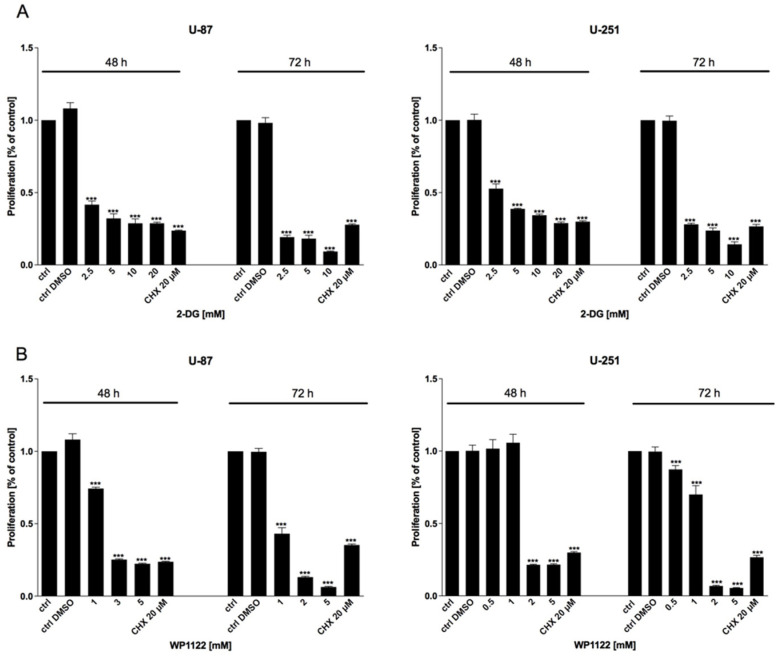
BrdU assay showing the proliferation of U-87 and U-251 cells after 48 and 72 h treatment with various concentrations of (**A**) 2-DG (0.5–20 mM) and (**B**) WP1122 (0.25–5 mM). As a positive control, CHX (20 μM) was used. Significant differences between the treatment means and control values are indicated by *** *p* < 0.001.

**Figure 3 biomedicines-09-01749-f003:**
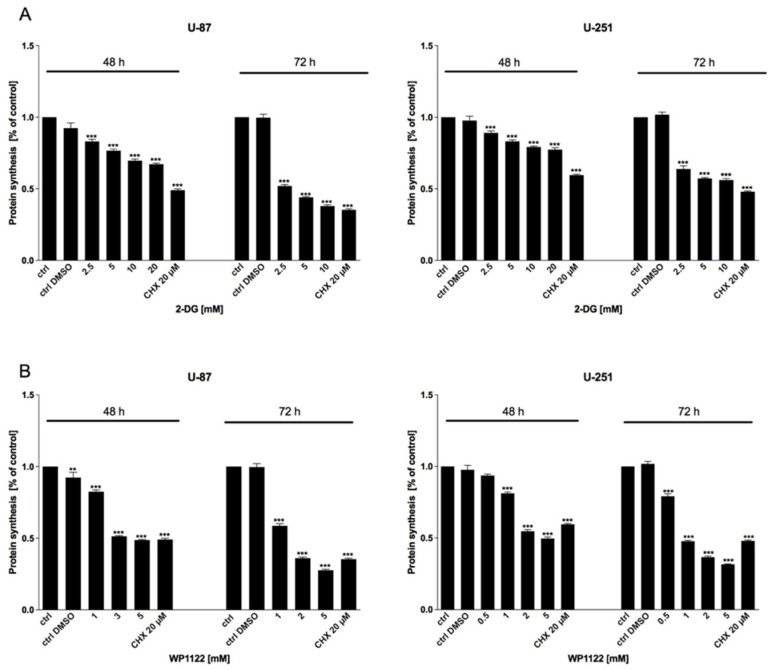
SRB assay showing protein synthesis of U-87 and U-251 cells after 48 and 72 h treatment with various concentrations of (**A**) 2-DG (0.5–20 mM) and (**B**) WP1122 (0.25–5 mM). As a positive control, CHX (20 μM) was used. Significant differences between the treatment means and control values are indicated by ** *p* < 0.01 and *** *p* < 0.001.

**Figure 4 biomedicines-09-01749-f004:**
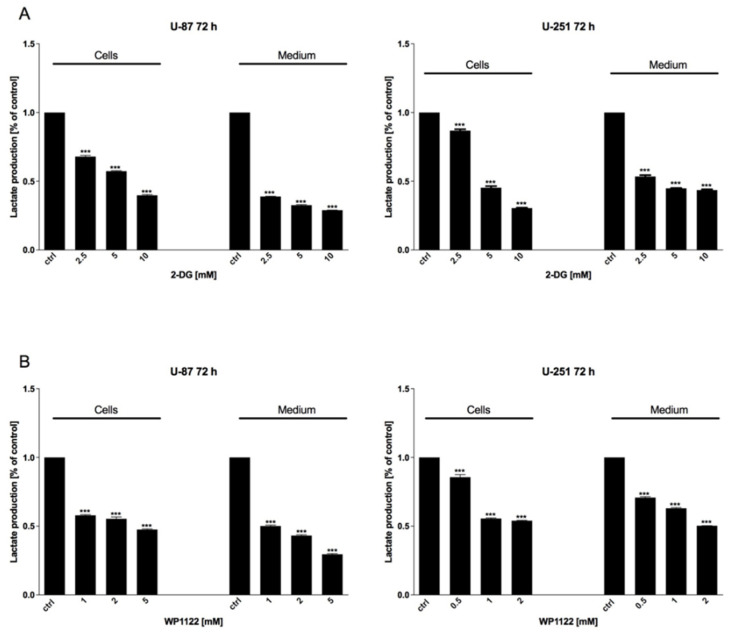
Lactate assay showing glycolytic activity and lactate synthesis of U-87 and U-251 cells after 72 h of treatment with various concentrations of (**A**) 2-DG (0.5–10 mM) and (**B**) WP1122 (0.25–5 mM). Significant differences between the treatment means and control values are indicated by *** *p* < 0.001.

**Figure 5 biomedicines-09-01749-f005:**
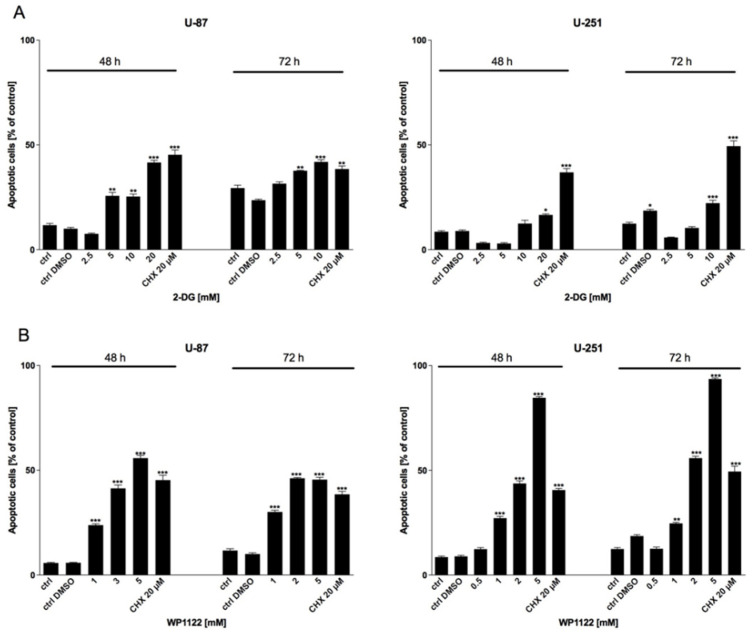
Apoptosis assay showing Annexin V-positive U-87 and U-251 cells after 48 and 72 h treatment with various concentrations of (**A**) 2-DG (0.5–20 mM) and (**B**) WP1122 (0.5–5 mM). As a positive control, CHX was used. Significant differences between the treatment means and control values are indicated by * *p* < 0.05, ** *p* < 0.01, and *** *p* < 0.001.

**Figure 6 biomedicines-09-01749-f006:**
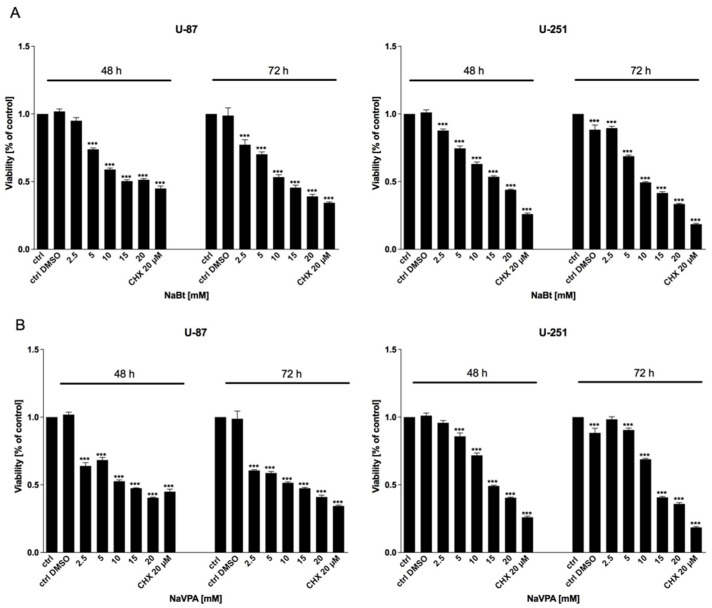
MTS assay showing the viability of U-87 and U-251 cells after 48 and 72 h of treatment with various concentrations of (**A**) NaBt (2.5–20 mM) and (**B**) NaVPA (2.5–10 mM). Significant differences between the treatment means and control values are indicated by *** *p* < 0.001.

**Figure 7 biomedicines-09-01749-f007:**
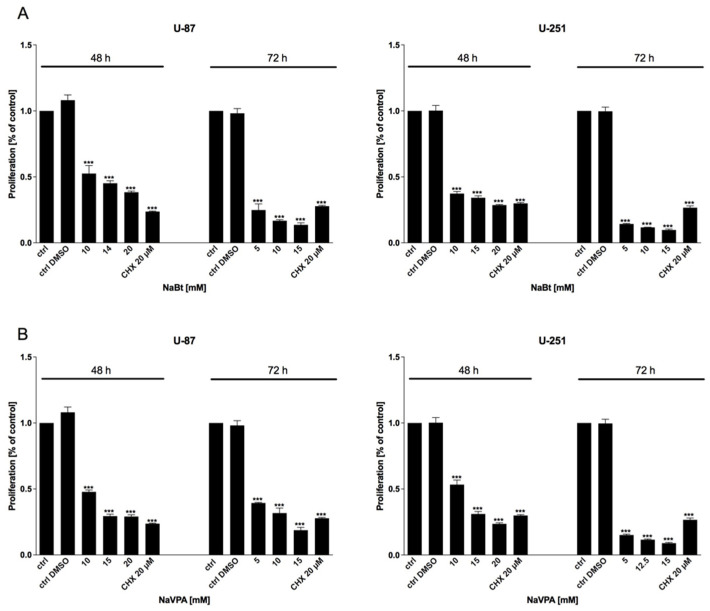
BrdU assay showing the proliferation of U-87 and U-251 cells after 48 and 72 h of treatment with various concentrations of (**A**) NaBt (2.5–20 mM) and (**B**) NaVPA (2.5–20 mM). Significant differences between the treatment means and control values are indicated by *** *p* < 0.001.

**Figure 8 biomedicines-09-01749-f008:**
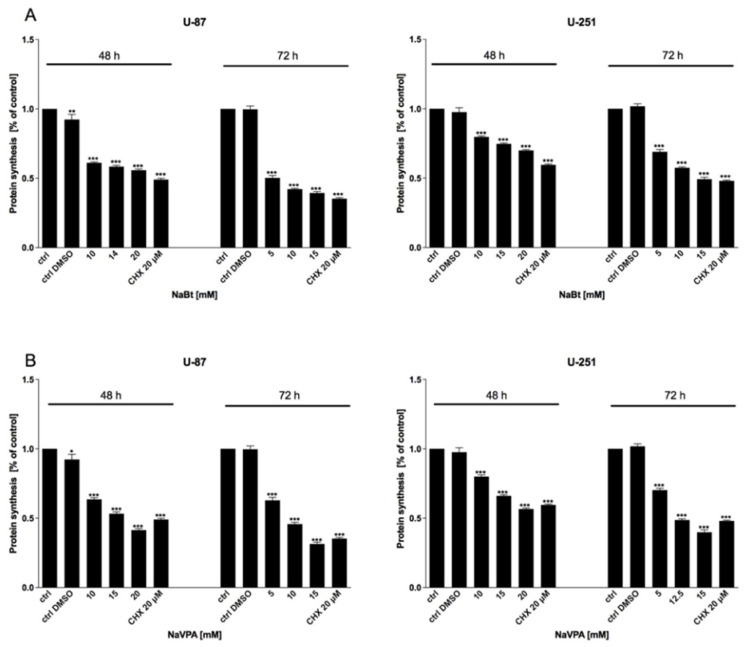
SRB assay showing protein synthesis of U-87 and U-251 cells after 48 and 72 h of treatment with various concentrations of (**A**) NaBt (2.5–20 mM) and (**B**) NaVPA (2.5–20 mM). Significant differences between the treatment means and control values are indicated by * *p* < 0.05, ** *p* < 0.01, and *** *p* < 0.001.

**Figure 9 biomedicines-09-01749-f009:**
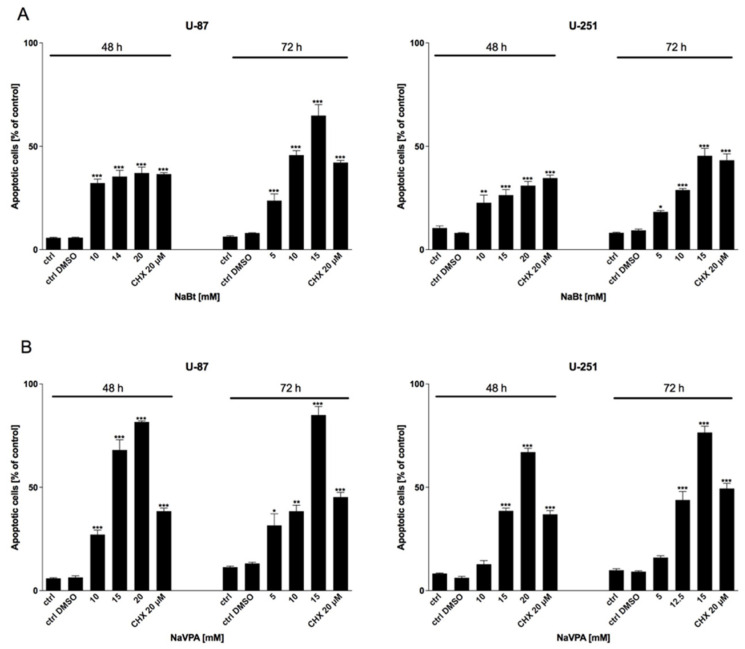
Apoptosis assay showing Annexin V-positive U-87 and U-251 cells after 48 and 72 h of treatment with various concentrations of (**A**) NaBt (5–20 mM) and (**B**) NaVPA (5–20 mM). As a positive control, CHX was used. Significant differences between the treatment means and control values are indicated by * *p* < 0.05, ** *p* < 0.01, and *** *p* < 0.001.

**Figure 10 biomedicines-09-01749-f010:**
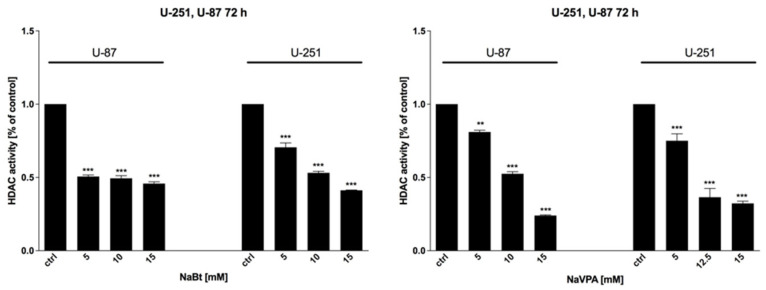
HDAC activity assay showing the inhibitory effect of NaBt (5–15 mM) and NaVPA (5–15 mM) treatment. Significant differences between the treatment means and control values are indicated by ** *p* < 0.01 and *** *p* < 0.001.

**Figure 11 biomedicines-09-01749-f011:**
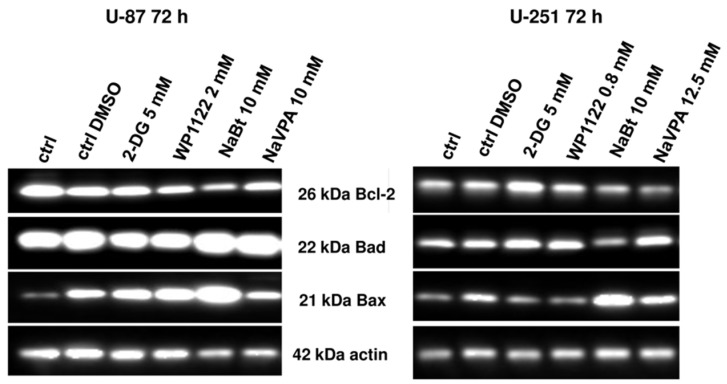
Representative blots showing the expression of antiapoptotic Bcl-2 and proapoptotic: Bad, Bax proteins after a IC50 concentration treatment with 2-DG (5 mM), WP1122 (2 mM for U87 and 0.8 mM for U-251 cells), NaBt (10 mM), and NaVPA (10 mM for U-87 and 12.5 mM for U-251 cells) (72 h). Actin was used as a loading control.

**Figure 12 biomedicines-09-01749-f012:**
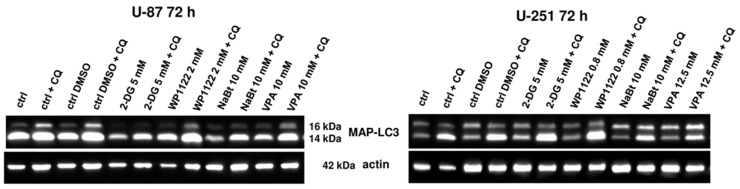
Representative blots showing the expression of autophagy marker, MAP-LC3 protein, after IC50 concentration treatment with 2-DG (5 mM), WP1122 (2 mM for U87 and 0.8 mM for U-251 cells), NaBt (10 mM), and NaVPA (10 mM for U-87 and 12.5 mM for U-251 cells) (72 h) alone or in combination with CQ (10 μM). Actin was used as a loading control.

**Figure 13 biomedicines-09-01749-f013:**
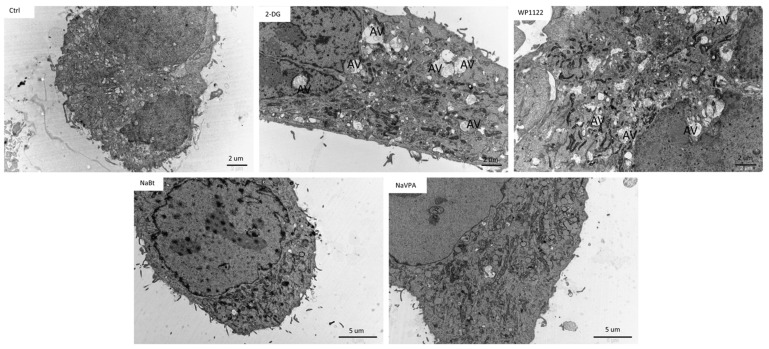
Transmission electron microscopy (TEM) images showing U-87 cell ultrastructure when treated with an IC50 concentration of 2-DG (5 mM), WP1122 (2 mM), NaBt (10 mM), and NaVPA (10 mM) (24 h). The presence of autophagic vacuoles is indicated as AV.

**Figure 14 biomedicines-09-01749-f014:**
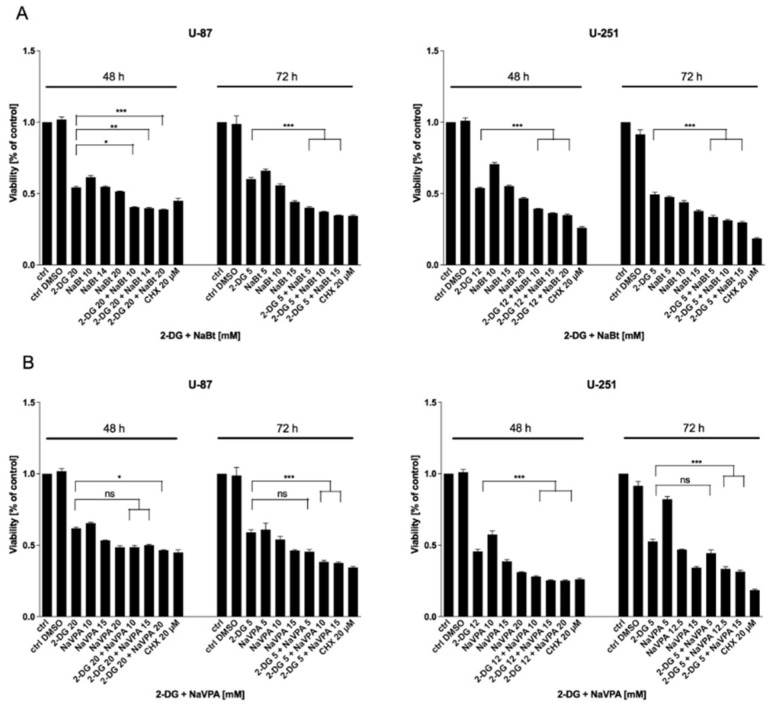
MTS assay showing the viability of U-87 and U-251 cells after 48 and 72 h of treatment with IC concentrations of 2-DG with concomitant presence of NaBt (**A**) or NaVPA (**B**), or WP1122 with concomitant presence of NaBt (**C**) or NaVPA (**D**). Significant differences between the treatment means and control values are indicated by ns-non-significant, * *p* < 0.05, ** *p* < 0.01, and *** *p* < 0.001.

## Data Availability

Not applicable.

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
