# Peer review of "Synergistic Anticancer Effect of Glycolysis and Histone Deacetylases Inhibitors in a Glioblastoma Model"

_biomedicines, 2021, doi:10.3390/biomedicines9121749_

Round 1
Reviewer 1 Report
In the work, „Synergistic anticancer effect of glycolysis and histone deacetylases inhibitors in glioblastoma model” the authors Beata Pająk et al. present a nice and informative in vitro study using two well established glioblastoma cell lines, in which the cytotoxic effects of the different glucose compounds are demonstrated, including details of protein biosynthesis inhibition and apoptosis. Further, the effects of histone deacetylases inhibitors, acting in concert with these, are shown.
The work is well done and stringent, however, some more general and some detailed points have to be mentioned.
First, the authors use long-term established, culture-adapted cell lines for their analysis. In particular regarding aerobic or anaerobic, metabolism, there may be differences between both the in vitro and the in vivo situation as well as between early-passage patient-derived cells and long-term cultivated cell lines. Second, by studying onls two lines, the value of the work is limited.
Some detailed aspects are relevant: as control, the authors always use the protein synthesis inhibitor cycloheximid. As such, cycloheximid is not a good apoptosis inductor, leding to low numbers of apoptotic cells in the related experiments.
Probably, not only the absolute concentration of deoxy-glucose compounds but the ratio of deoxy-glucose to glucose is relevant for the biological effects. This should be considered, in particular for possible clinical application; are there animal models?
In fig. 11, the actin control looks a bit uneven.
The effects of chloroquine on the experiments shown in figures 1-4 would also be interesting (maybe it is in fig. S4).
The discussion section is a bit long.
Unfortunately, the language of the work is imperfect in some sentences, and would merit a workover by a native speaker.
Author Response
In the work, „Synergistic anticancer effect of glycolysis and histone deacetylases inhibitors in glioblastoma model” the authors Beata Pająk et al. present a nice and informative in vitro study using two well-established glioblastoma cell lines, in which the cytotoxic effects of the different glucose compounds are demonstrated, including details of protein biosynthesis inhibition and apoptosis. Further, the effects of histone deacetylases inhibitors, acting in concert with these, are shown. The work is well done and stringent, however, some more general and some detailed points have to be mentioned.
- Thank you for the positive comment and appreciation of our efforts. We tried to introduce all possible for us experimental tools to verify our hypothesis. Please, find below our responses to Your comments.
First, the authors use long-term established, culture-adapted cell lines for their analysis. In particular regarding aerobic or anaerobic, metabolism, there may be differences between both the in vitro and the in vivo situation as well as between early-passage patient-derived cells and long-term cultivated cell lines. Second, by studying only two lines, the value of the work is limited.
- We fully agree that using commercial cell lines as an experimental model has its limitations and all in vitro results should be further confirmed with in vivo models. We added a critical comment to the Discussion section to underline the importance of further verifying the obtained results.
Lines 1725-1726 „As far as we know, this is the first evidence that targeting GBM metabolism along with genes transcription processes is an effective anticancer strategy. It should be underlined that presented results are obtained from in vitro studies and should be further confirmed in in vivo models. We are also aware that clinical approval of combined therapy is more time- and money-consuming”.
We want to explain, however, that numerous papers are describing preclinical and clinical effects of 2-DG, that were recently summarized by our team in a review paper: Pajak, B.; Siwiak, E.; Sołtyka, M.; Priebe, A.; Zieliński, R., Fokt, I.; Ziemniak, M.; Jaśkiewicz, A.; Borowski, R.; Domoradzki, T., Priebe, W. 2-deoxy-D-glucose and its analogs: from diagnostic to therapeutic agents. Int J Mol Sci. 2020, 21, 234. On the other hand, WP1122 has recently been approved by MHRA and ethical authorities in the UK as a drug candidate for the phase1 clinical studies in COVID-19 patients (due to the upregulated glycolysis in SARS-CoV-2 viral infection) (https://www.pharmaceutical-business-review.com/news/moleculin-mhra-phase-ia-wp1122-covid-19/). There are already available in vivo pharmacokinetics, toxicology, and proof-of-concept studies from animal GBM and pancreatic cancer models submitted to the agency MHRA, showing high anti-cancer efficiency and drug safety. These analyses were performed via MD Anderson Cancer Center, thus could not be a part of our manuscript focused on new possible treatment directions in a combination treatment.
Some detailed aspects are relevant: as control, the authors always use the protein synthesis inhibitor cycloheximide. As such, cycloheximide is not a good apoptosis inductor, leading to low numbers of apoptotic cells in the related experiments.
- Cycloheximide is a widely known apoptosis inducer that was used in our previously published studies and is also cited by author authors (FEBS Lett 1997, 1, 113-116; Plos One 2016, doi.org/10.1371/journal.pone.0164003). We agree that CHX-treatment could induce cell detachment and cell number reduction, thus whenever the cells were collected for analysis, medium from CHX-treated cells was collected and centrifuged and scraped cells, as well as centrifuged apoptotic cells were analyzed. The additional information has been added to the Material and Method section.
Lines 252-253: “Due to the CHX-induced cell detachment and possible cell number reduction, medium from CHX-treated cells have been collected and centrifuged and scraped, as well as retrieved cells were analyzed”.
Probably, not only the absolute concentration of deoxy-glucose compounds but the ratio of deoxy-glucose to glucose is relevant for the biological effects. This should be considered, in particular for possible clinical application; are there animal models?
- It is a significant comment, and we fully agree with the Reviewer that the use of 2-DG is limited due to its competition with glucose present in the blood. Both compounds use the same GLUT transporters to enter the cells. Thus, it is one of the major limitations of 2-DG clinical use. This point has been addressed in the Discussion section:
Lines 1654-1655: “Despite the numerous preclinical and clinical studies, the use of 2-DG in anticancer therapy is still limited. Its rapid metabolism and short half-life (according to Hansen et al., after infusion of 50 mg/kg 2-DG, its plasma half-life was only 48 min. [13] make 2-DG a rather poor drug candidate. Moreover, 2-DG has to be used at relatively high concentration ( 5 mmol/L) in order to compete with blood glucose”.
The presence of glucose is not the critical factor for WP1122 action, due to its lipophilic properties related to the presence of acetyl groups. It has been clarified in the Discussion section.
Line 1661: “Due to its lipophilic properties related to the presence of acetyl groups, WP1122 enters cells and, importantly, crosses the BBB by passive diffusion rather than relying upon a specific glucose transporter”.
In fig. 11, the actin control looks a bit uneven.
- We agree with the Reviewer that there is a slight difference in the actin level in the presented blot. We added an additional Table (Table S1) to the Supplementary Materials with the densitometric result.
The effects of chloroquine on the experiments shown in figures 1-4 would also be interesting (maybe it is in fig. S4).
- The effect of CQ on the viability of cells treated with IC50 concentration of all tested compounds is presented in Figure S4B. Because there were no significant differences between the viability of compound/compound + CQ treatments, we did not examine further other cytotoxicity assays that were expected to correspond with MTS. It could be possible that autophagy is a primary response to glycolysis inhibition, and evaluation of the CQ effect after 72 h could be too long. We addressed this point additionally in the Results section.
Line 1339-1341: “On the other hand, in NaBt [10 mM]- and NaVPA [10 mM]-treated cells, autophagolysosomes were not visible. It suggests that glycolysis inhibition can modulate the autophagy process, but its downstream cellular effects and lack of ATP synthesis strongly induce autophagy-independent cell death. It is also possible that autophagy is a primary response to glycolysis inhibition and is activated very early, thus was observed with TEM analysis after 12 h of incubation, whereas viability was assessed after 48 and 72 h of treatment”.
The discussion section is a bit long.
- We shorten the Discussion according to the Reviewer’s suggestions, and some parts have been removed.
Unfortunately, the language of the work is imperfect in some sentences and would merit a workover by a native speaker.
- To improve the language, the manuscript has been corrected by MDPI English Editing Service.
Reviewer 2 Report
Dear Sirs
You have a wonderful story that deserves to be published but there are few things to prevent it from acceptance. First of all, I did not see negative control such as normal cells or astrocytes. I ,honestly, will not buy into possible argument that you conducted that testing before with described drugs or this can be of future plans. At least for some of presented study these controls should be presented. Secondly, since you do not provide in vivo effect using ic glioblastoma model and mice survival in the drug presence, the study of yours requires testing you hypothesis using primary GBM. This also a must to publish your story to demonstrate your effect of drug combination using clinical relevant cell culture ( up to 5 passage). Finally, I would like to see 2-DG analog uptake by GBM cells during blocking GLUT1 and GLUT2 via immunofluorescense.
Author Response
You have a wonderful story that deserves to be published but there are few things to prevent it from acceptance. First of all, I did not see negative control such as normal cells or astrocytes. I , honestly, will not buy into possible argument that you conducted that testing before with described drugs or this can be of future plans. At least for some of presented study these controls should be presented. Secondly, since you do not provide in vivo effect using ic glioblastoma model and mice survival in the drug presence, the study of yours requires testing you hypothesis using primary GBM. This also a must to publish your story to demonstrate your effect of drug combination using clinical relevant cell culture (up to 5 passage). Finally, I would like to see 2-DG analog uptake by GBM cells during blocking GLUT1 and GLUT2 via immunofluorescense.
- We acknowledge the Reviewer’s comments, and we agree that the in vitro analysis are the first step of drug development, that have to be further verified using in vivo models. We addressed this point in the Discussion section.
Lines 1741-1742 „As far as we know, this is the first evidence that targeting GBM metabolism along with genes transcription processes is an effective anticancer strategy. It should be underlined that presented results are obtained from in vitro studies and should be further confirmed in in vivo models.
I would like to assure the Reviewer that we do not declare statements concerning our drugs that are not supported by already known data. Our declaration was related to new 2-DG analog, WP1234 that is currently investigated in our lab, and we are planning to submit the manuscript showing its efficiency till the end of this year. With regards drugs safety of animal model, we want to explain, however, that numerous papers are describing preclinical and clinical effects of 2-DG, that were recently summarized by our team in a review paper: Pajak, B.; Siwiak, E.; Sołtyka, M.; Priebe, A.; Zieliński, R., Fokt, I.; Ziemniak, M.; Jaśkiewicz, A.; Borowski, R.; Domoradzki, T., Priebe, W. 2-deoxy-D-glucose and its analogs: from diagnostic to therapeutic agents. Int J Mol Sci. 2020, 21, 234. On the other hand, WP1122 has recently been approved by MHRA and ethical authorities in the UK as a drug candidate for the phase1 clinical studies in COVID-19 patients (due to the upregulated glycolysis in SARS-CoV-2 viral infection) (https://www.pharmaceutical-business-review.com/news/moleculin-mhra-phase-ia-wp1122-covid-19/). There are already available in vivo pharmacokinetics, toxicology, and proof-of-concept studies from animal GBM and pancreatic cancer models submitted to the agency MHRA, showing high anti-cancer efficiency and drug safety. The toxicology studies in mice proved that WP1122 did not exert any negative effects on normal tissue up to 9 g/kg m.c., whereas the effective anticancer concentrations were 0.5-2 g/kg m.c. Thus, we did not include normal cells in our model. These analyses were performed via MD Anderson Cancer Center, thus could not be a part of our manuscript focused on new possible treatment directions in a combination treatment. However, there are some open access results:
In the case of our manuscript, we verified hypothesis whether the concomitant inhibition of glycolysis and HDAC could be a new strategy for GBM treatment. We confirmed that there is a synergistic effect thus we are working with WP1234. After in vitro experiments, WP1234 will be tested in iv vivo models.
With regard to intracellular uptake of WP1122, I would like to explain that our Prof. Waldemar Pribe’s team examined the intracellular uptake of labeled 2-DG and WP1122:
EXTH-07. DESIGN AND EVALUATION OF WP1122, AN INHIBITOR OF GLYCOLYSIS WITH INCREASED CNS UPTAKE.
Waldemar Priebe, Rafal Zielinski, Izabela Fokt, Edward Felix, Venugopal Radjendirane, Jayakumar Arumugam, Matthew Tai Khuong, Maciej Krasinski, Stanislaw Skora
Neuro-Oncology, Volume 20, Issue suppl_6, November 2018, Page vi86, https://doi.org/10.1093/neuonc/noy148.356
Published:
05 November 2018
Our in vitro experiments confirmed inhibition of glycolysis in U87 cells and high sensitivity of broad spectrum of cancer cells to WP1122 both in hypoxic and normoxic conditions (IC50 range from 1–10 mM). In vivo studies showed that WP1122 is well tolerated by animals even with prolonged exposure and extend survival of mice in orthotopic U87 GBM model. Initial pharmacokinetic experiments demonstrated rapid uptake of WP1122 after oral administration allowing to achieve two orders of magnitude higher maximum concentration of 2-DG in plasma, compared to animals treated with an equal molar dose of pure 2-DG. We also observed significantly higher levels of 2-DG in brains of mice treated with WP1122 than in mice receiving equal dose of 2-DG. In summary, WP1122 is a biologically effective prodrug of 2-DG with a good toxicity profile and promising pharmacokinetic characteristics that warrants detail preclinical and clinical development as a potential therapeutic agent for glioblastomas and other highly glycolytic tumors.
Unfortunately, the WP1122 and 2-DG uptake have not been tested in GLUT-knockout cells. There were some attempts to perform such experiments. However, maintaining cell culture survival without the ability of the cells to uptake glucose from the medium resulted in cell death and obstacles with the generation of a reliable cellular model. Thus, the PK/PD analysis in animals has been performed.
Author Response :
You have a wonderful story that deserves to be published but there are few things to prevent it
from acceptance. First of all, I did not see negative control such as normal cells or astrocytes. I ,
honestly, will not buy into possible argument that you conducted that testing before with
described drugs or this can be of future plans. At least for some of presented study these controls
should be presented. Secondly, since you do not provide in vivo effect using ic glioblastoma
model and mice survival in the drug presence, the study of yours requires testing you hypothesis
using primary GBM. This also a must to publish your story to demonstrate your effect of drug
combination using clinical relevant cell culture (up to 5 passage). Finally, I would like to see 2-DG
analog uptake by GBM cells during blocking GLUT1 and GLUT2 via immunofluorescense.
- We acknowledge the Reviewer’s comments, and we agree that the in vitro analysis are
the first step of drug development, that have to be further verified using in vivo models.
We addressed this point in the Discussion section.
Lines 1741-1742 „As far as we know, this is the first evidence that targeting GBM
metabolism along with genes transcription processes is an effective anticancer strategy. It
should be underlined that presented results are obtained from in vitro studies and should be
further confirmed in in vivo models.
I would like to assure the Reviewer that we do not declare statements concerning our
drugs that are not supported by already known data. Our declaration was related to new
2-DG analog, WP1234 that is currently investigated in our lab, and we are planning to
submit the manuscript showing its efficiency till the end of this year. With regards drugs
safety of animal model, we want to explain, however, that numerous papers are
describing preclinical and clinical effects of 2-DG, that were recently summarized by our
team in a review paper: Pajak, B.; Siwiak, E.; Sołtyka, M.; Priebe, A.; Zieliński, R., Fokt, I.;
Ziemniak, M.; Jaśkiewicz, A.; Borowski, R.; Domoradzki, T., Priebe, W. 2-deoxy-Dglucose
and its analogs: from diagnostic to therapeutic agents. Int J Mol Sci. 2020, 21, 234.
On the other hand, WP1122 has recently been approved by MHRA and ethical authorities
in the UK as a drug candidate for the phase1 clinical studies in COVID-19 patients (due to
the upregulated glycolysis in SARS-CoV-2 viral infection) (https://www.pharmaceuticalbusiness-
review.com/news/moleculin-mhra-phase-ia-wp1122-covid-19/). There are
already available in vivo pharmacokinetics, toxicology, and proof-of-concept studies
from animal GBM and pancreatic cancer models submitted to the agency MHRA,
showing high anti-cancer efficiency and drug safety. The toxicology studies in mice
proved that WP1122 did not exert any negative effects on normal tissue up to 9 g/kg m.c.,
whereas the effective anticancer concentrations were 0.5-2 g/kg m.c. Thus, we did not
include normal cells in our model. These analyses were performed via MD Anderson
Cancer Center, thus could not be a part of our manuscript focused on new possible
treatment directions in a combination treatment. However, there are some open access
results:
In the case of our manuscript, we verified hypothesis whether the concomitant inhibition
of glycolysis and HDAC could be a new strategy for GBM treatment. We confirmed that
there is a synergistic effect thus we are working with WP1234. After in vitro experiments,
WP1234 will be tested in iv vivo models.
With regard to intracellular uptake of WP1122, I would like to explain that our Prof.
Waldemar Pribe’s team examined the intracellular uptake of labeled 2-DG and WP1122:
EXTH-07. DESIGN AND EVALUATION OF WP1122, AN INHIBITOR OF
GLYCOLYSIS WITH INCREASED CNS UPTAKE.
Waldemar Priebe, Rafal Zielinski, Izabela Fokt, Edward Felix, Venugopal
Radjendirane, Jayakumar Arumugam, Matthew Tai Khuong, Maciej Krasinski,
Stanislaw Skora
Neuro-Oncology, Volume 20, Issue suppl_6, November 2018, Page vi86,
https://doi.org/10.1093/neuonc/noy148.356
Published:
05 November 2018
Our in vitro experiments confirmed inhibition of glycolysis in U87 cells and high
sensitivity of broad spectrum of cancer cells to WP1122 both in hypoxic and normoxic
conditions (IC50 range from 1–10 mM). In vivo studies showed that WP1122 is well
tolerated by animals even with prolonged exposure and extend survival of mice in
orthotopic U87 GBM model. Initial pharmacokinetic experiments demonstrated rapid
uptake of WP1122 after oral administration allowing to achieve two orders of magnitude
higher maximum concentration of 2-DG in plasma, compared to animals treated with an
equal molar dose of pure 2-DG. We also observed significantly higher levels of 2-DG in
brains of mice treated with WP1122 than in mice receiving equal dose of 2-DG. In
summary, WP1122 is a biologically effective prodrug of 2-DG with a good toxicity profile
and promising pharmacokinetic characteristics that warrants detail preclinical and
clinical development as a potential therapeutic agent for glioblastomas and other highly
glycolytic tumors.
Unfortunately, the WP1122 and 2-DG uptake have not been tested in GLUT-knockout cells.
There were some attempts to perform such experiments. However, maintaining cell culture
survival without the ability of the cells to uptake glucose from the medium resulted in cell
death and obstacles with the generation of a reliable cellular model. Thus, the PK/PD analysis
in animals has been performed.
Round 2
Reviewer 2 Report
all comments are being addressed